# Tennis Attack: An Exergame Utilizing a Natural User Interface to Measure and Improve the Simple Reaction Time

**Nikolaos Politopoulos**  **and Thrasyvoulos Tsiatsos \***

Department of Informatics, Aristotle University of Thessaloniki, 54124 Thessaloniki, Greece
\* Correspondence: tsiatsos@csd.auth.gr

**Abstract:** The present article demonstrates the process of designing, developing, implementing, and evaluating an active video game or exergame. The main goal of our proposed exergame is to develop the simple reaction time of players. The main target group are simple users; however, it can work on tennis specialists. Herein, we used the exergame to investigate the hypothesis that players can improve their reaction time through practice. To achieve this, players' simple reaction time was measured at the start of the game. Then, the players took part in a 4-week training session. At the end of the training session, the simple reaction time of players was measured again and a questionnaire was completed. Another goal of this paper was to investigate the difference between perceived usefulness (general usability, usefulness, and user interface satisfaction) of experts and non-experts (sports science students and computer science students) and pro-gamers and casual gamers. The findings were encouraging. The majority of the players reported that the gaming experience was very satisfying and the game was easy to use and learn. Moreover, after the analysis, it was discovered that this game can significantly improve the simple reaction time of all players. This improvement was independent from the players' background.

**Keywords:** active video game; natural user interfaces; Microsoft Kinect; simple reaction time (SRT)



## 1. Introduction

Nowadays, people enjoy playing video games in their daily life, in various ways—they play them on consoles, computers, and smartphones. In fact, video games were introduced in the late 50s, but they became more popular in the early 70s. Children feel enthusiastic while playing video games and consequently, parents, teachers, and psychologists are deeply concerned. They assume that kids' mania for video games will make them unsocial and unenthusiastic in every other aspect of their life, such as school or sports. It is a firm fact that video games are a social phenomenon and one of the most profitable industries of the 21st century [1].

In the last decade, game developers have created video games that support exercise and healthy lifestyles. This genre of games is called active video games (AVGs) or "exergames". Experts consider these games as important tools that can transform a traditional sedentary activity, such as playing video games, into physical exercise [2].

To achieve this, game developers use natural user interfaces (NUIs). Trying to define NUIs is not easy, but frequently, when we try to consider user interfaces that are both natural and easy to use, we think of user interfaces where the interaction is direct and consistent with our "natural" behavior. Examples of user interfaces that people frequently consider as natural are the presence of more than one point of contact or multi-touch on the Apple iPad or using body gestures to control Microsoft's Kinect console. Experts believe that NUIs are the next step in the evolution line of user interfaces. The benefit of NUIs is that the user interaction feels direct, easy, fun, and natural since the user can use a variety of basic skills in comparison with a more conventional graphical user interface interaction—which mostly occurs through a mouse and keyboard.

NUIs utilize skills that we have acquired and developed through a lifetime of living. As a result, the cognitive load and distractions are minimized. In addition, it is important to design NUIs with the use context in mind, since no user interface can be natural in all use cases and to all users. While gestures, voice, and touch are important components of many NUIs, they will only feel natural to a user if they match their skill level and use context.

Moreover, NUIs should take advantage of the potential users' skills, since users want to avoid the trouble of learning something completely new. In this case, users can apply the skills they have developed from their daily life, both in their free time and working environment. Once the users understand which skills are needed, they can apply their existing skills and expectations to interact with the NUIs. There are two potential ways of creating this:

- Reusing common human skills;
- Reusing domain-specific skills.

In an attempt to determine common human skills, we refer to the things that most people know how to do (e.g., talking). On the contrary, domain-specific skills are determined as skills of a particular user group (e.g., designing NUIs for doctors). NUIs should fit the individual user and his use context in order that they feel natural to him. In [3], we stated four guidelines that we should consider while designing NUIs:

- Instant expertise;
- Progressive learning;
- Direct interaction;
- Cognitive load (primarily using innate abilities and simple skills).

Graf [4] conducted a study to assess and set side-by-side the possible rates of energy consumption and the related physiologic responses in children while playing active video games. The authors used Wii Sports (https://www.nintendo.com/wiifit/launch/wiifitplus/ (accessed on 18 September 2022)) and Dance Dance Revolution (DDR) (https://www.ddrgame.com/ (accessed on 18 September 2022)). The findings indicated that energy consumption, heart rate, and perceived exertion of both Wii Sports and DDR games are similar or even higher than moderate-intensity walking (4.2–5.7 km/h). Additionally, LeBlanc [5] reported the results of controlled studies that show AVGs slightly increase physical activity. However, the results regarding whether AVGs lead to decreases in stationary behavior are less clear. AVGs seem to provide some health benefits in special populations, but there is not enough data to recommend AVGs as a standard procedure to increase daily physical activity. This finding triggered the current study which focuses on human reaction time and its improvement with an AVG.

On the contrary, Pedersen [6] stated that using a commercial natural user interface as input device (e.g., Nintendo Wii) might not be as effective to improve motor skills in children compared to more traditional physical education. However, Benzing [7] conducted research on children with attention-deficit/hyperactivity disorder (ADHD) and the findings were impressive. They found that exergaming can be beneficial to children's motor skills and motivates children to be involved. Moreover, Hilton [8] found that exergames can be extremely useful for children with autism.

Another research from Zeng [9] stated that exergames have the potential to lessen weight gain for overweight or/and obese children and youth. Exergaming seems to motivate and engage children in physical activity. As a result, exergaming may enable children and youth to follow the standard guidelines for 1 h of moderate to intense exercise per day. In other words, exergaming shows promise as a tool to motivate and engage children as well as adolescents in physical activity.

Page [10] reviewed the possibility that active video games improve motor skills on both children and young adults. In addition, there was strong evidence that active video games improve balance. Moreover, these games appeared to be beneficial to participants with Cerebral Palsy. The authors stated that active video games could be a remarkable tool to improve gross motor skills of non-typically developing children. Furthermore,

Hocking [11] reviewed whether active video games improve motor function in people with developmental disabilities. The results proved that they are quite effective in growing motor skills, but their effectiveness depends on training intensity.

Simple reaction time (SRT) is defined, for a person or system, as the time interval between a given stimulus or event and the response [12]. Niemi and Näätänen [13] stated that reaction time measures are used for two reasons: (a) To study the mental process by measuring the time it takes to perform a certain process or a part of it, and (b) to study the reaction process by continuously changing the stimuli and responses. Additionally, the authors measured whether the fore period to reaction time is crucial to people's reaction time. The fore period is the time of reference in which a subject prepares to respond to the stimuli on a certain trial. The results proved that subjects cannot perceive the moment when they started preparing their reaction. Therefore, subjects have to be alert at a high degree of motor preparation to be able to quickly respond when the stimulus is delivered. These findings were crucial for us to design our game.

The present paper presents a study of how active video games can improve players' simple reaction time with NUIs. Herein, Section 2 presents the related works, explaining the improvement in reaction time and types of technologies that have been used. Section 3 introduces the research questions of the study and Section 4 demonstrates the materials used, the participants, the instruments, and the methodology followed to carry out our research. Section 5 presents the results and evaluation activities and Section 6 provides a discussion of these results. Finally, Section 7 concludes our remarks and presents the future steps.

## 2. Related Works

Several studies demonstrate the correlation between gamers and SRT. Unfortunately, there are a few games that use NUIs to improve SRT. Researchers usually use casual games to examine this hypothesis.

A gamer is someone who plays various types of video games online or as a single player and considers it a hobby. Generally, a gamer refers to any kind of gaming enthusiast; however, when used in Information Technology, the term refers to those that utilize a full range of electronic or digital games.

Deleuze [14] conducted a study to measure the reaction time between players of different genres. In general, the results showed that gamers increase their reaction time, but that increase differs between genres. Another study conducted by Dye [15] measured the possible difference in reaction time of active video games between casual, not regular, regular, and professional gamers. Casual gamers are defined as players that play games from time to time, not regular gamers are players that play games two to three times per week, regular gamers are players that play games every day, and professional gamers are players that play games for a living, in tournaments or in stream services. The results were promising since professional gamers reacted faster and had a smaller reaction time compared to casual gamers. Additionally, the authors found that the professional gamers' reactions were not as impulsive as casual gamers, who were anticipating the next step of the game.

Guzman et al. [16] conducted research to measure the acute effects of exercise and active video games on adults' reaction time and their perceived exertion. The results showed that aerobic exercise combined with AVGs improved players' SRT compared to aerobic exercise without AVGs. On the contrary, the results failed to support that aerobic exercise performed in combination with AVGs would improve the complex reaction time more than only aerobic exercise. Moreover, Franceschini et al. [17] conducted research on children with dyslexia using active video games. The results here were impressive. Children with dyslexia improved their reaction time as well as their reaction to the stimuli. Additionally, the children improved their visual attention, which is very important for orthographic transparency.

Stroud et al. [18] conducted three experiments and tested the attentional components of two groups (younger and older adults) by generally playing three different casual games. The findings suggest that casually playing video games improved the reaction time. Additionally, Anderson et al. [19] found that playing video games aggressively and with passion increases visual motor skills as well as the reaction time of players.

Dorval et al. [20] and Nielsen et al. [21] stated that playing video games can induce beneficial effects, including increased performance on eye-hand coordination tasks and neuropsychological tests, as well as better reaction time, spatial visualization, and mental rotation. Meanwhile, Green et al. assessed the enhancement of video games on visual attention. Moreover, Bavelier et al. [22] stated that players who play action games have a 10% faster reaction time than other players.

Eichenbaum et al. [23] conducted research on adults with amblyopia. The authors closed the "good" eye with an eye-patch and allowed the adults to play an action video game. The improvement in their visual motor skills was dramatically better than the control group who was only engaged with simple activities, such as knitting and reading.

Glueck et al. [24] conducted research to investigate the hypothesis that a mixed reality action game could improve SRT and balance. The results proved that after a training period of 8 h, there was a statistically significant improvement in the reaction time of players.

Hulteen et al. [25] conducted research to investigate whether training motor skills with AVGs can transform this ability into the real world. The results proved that some motor skills are better trained through an AVG, while others need some improvement.

The present research is an attempt to combine the use of pre- and post-examination along with in-game measurements to reveal the possibilities of improving SRT using an active video game based on NUIs.

## 3. Research Questions

As previously mentioned, the present study is an attempt to investigate the improvement in SRT by exploiting an active video game based on NUIs. To achieve this, additional issues need to be investigated, except for the improvement in SRT. The main issues to explore are:

(a)   The effect of Tennis Attack on players' SRT during and after the gameplay. Here, in-game SRT is defined as the time of the scores that each player will achieve while playing the game and real-world SRT as the time of the scores that each player will achieve while taking the validated test after playing the game;

(b)   the effect of the game on different types of players based on their expertise or as gamers;

(c)   the way that players adapt to Tennis Attack.

With this aim, the research questions are as follows:

- RQ1: Does Tennis Attack improve the in-game reaction time of players?
- RQ2: Does Tennis Attack improve the real-world reaction time of players?
- RQ3: Is there any significant relationship between a player's expertise as an athlete and their improvement?
- RQ4: Is there any significant relationship between a player's gaming experience and their improvement?
- RQ5: Is Tennis Attack user friendly to all users?

## 4. Method

This section presents the methodology for obtaining responses to our research questions.

### 4.1. Participants

The study involved 60 undergraduate and postgraduate students (31 male and 29 female, M = 22.57 years of age, SD = 1.88) chosen randomly from the Department of Physical Education and Sport Science (PESS) and Computer Science (CS) Department. The participants

consisted of 35 CS students and 25 PESS students. Among the 60 participants, 24 declared that they are experienced/pro-gamers who played for more than 5 h per day. Before the assessment activity, all of the participants were asked to fill an online demographical data questionnaire, which included questions regarding the students' age, frequency of computer use, and gaming experience. Additionally, 26 of the participants were Tennis athletes. Notably, this research is part of a PhD study and the present methodology has been reviewed by the competent research committee.

The results indicated that the students used a computer on a regular basis and most of them had previous experience in video games and Microsoft Kinect.

### *4.2. Instruments*

The instruments used to answer the research questions will be presented in the following sections.

### 4.2.1. USE Questionnaire

To evaluate the usability of the game, players were asked to complete the USE questionnaire [26] after the activity was concluded. The questionnaire is designed to effectively measure the most important aspects of a product's usability. It consists of 30 questions grouped into height dimensions: (a) Usefulness, (b) Ease of Use, (c) Ease of Learning, and (d) Satisfaction. A 7-point Likert rating scale was employed with the following anchors: 1 strongly disagree, 2 disagree, 3 somewhat disagree, 4 neutral, 5 somewhat agree, 6 agree, and 7 strongly agree.

### 4.2.2. Reaction Time Test

To measure the reaction time of users, we used a simple verified online click test by Human Benchmark [27]. This test was proposed by experts for its ease of use and ease of accurately measuring the SRT. Each user sat in front of a computer and had to click the left key of the mouse when the light turned from red to green.

### *4.3. Tennis Attack Serious Game*

Tennis Attack is a serious AVG that utilizes the depth camera of Microsoft Kinect as a controller. The aim of this game is to improve the SRT of both athletes and casual users.

The general idea of the game is based on the BATAK system (Figure 1) [28]. BATAK is a training system that uses a set of lights on a wall. The trainer must stand in front of this system and react with the lights when they are turned on. The trainer must hit them when they are turned on and switch them off. It is a system designed to measure and improve athletes' reaction time and is commonly used as part of Formula 1 drivers' training program. Additionally, it was decided that the players will not receive any feedback directly from the game during the training session. Therefore, they will not have any reinforcement neither positive nor negative. At the end of their session, the players can monitor their scores.

Tennis Attack was developed using the Unity game engine and Zigfu Development Kit or "ZDK". The game can be considered as a tool to measure and improve SRT (one stimulus—one response). It is a game that can be configured by the players and allows them to control their training session by deciding the number of balls and the rate at which they will appear on screen.

All data are stored in a database at a local server, which allows the users to monitor their performance and observe their progress.

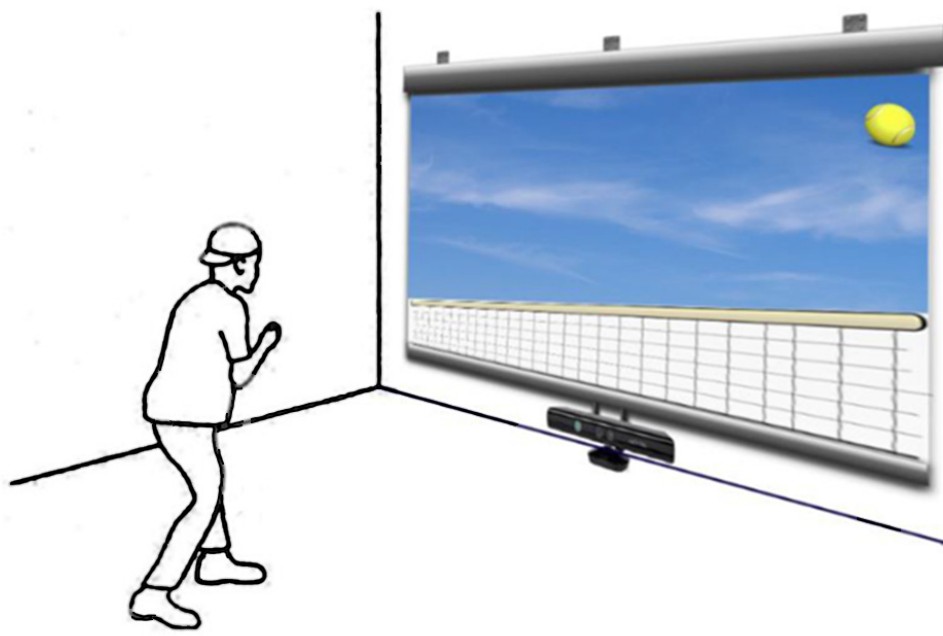

**Figure 1.** Tennis Attack wireframe.

*4.4. Procedure*

The experiment procedure consisted of three main phases: (1) Measuring the reaction time before the training session, (2) training session, (3) measuring the reaction time after the training session.

First, in phase 1, each participant took a validated test to measure and record their reaction time. Then, in phase 2, the duration of the training session was 4 weeks. Every participant had to play the game three times per week. The total sessions per player were twelve. The play time was divided into five different rounds. Each round lasted for 1 min, and every player had to respond to 30 balls. The interval between each ball was 2 s (Figure 2).

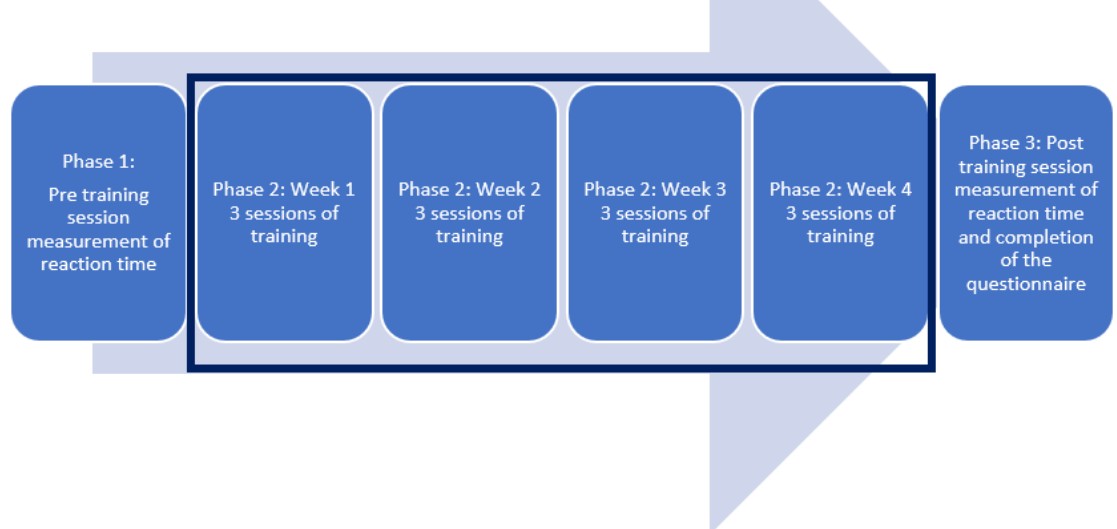

**Figure 2.** Experiment procedure.

At the beginning and at the end of these two phases, each participant will play a round to measure and record their reaction time (Figure 3). This procedure is a standard training procedure for Tennis players to simulate Tennis Ball machines. Finally, in phase 3 and similar to phase 1, each participant took a validated test to measure and record their reaction time.

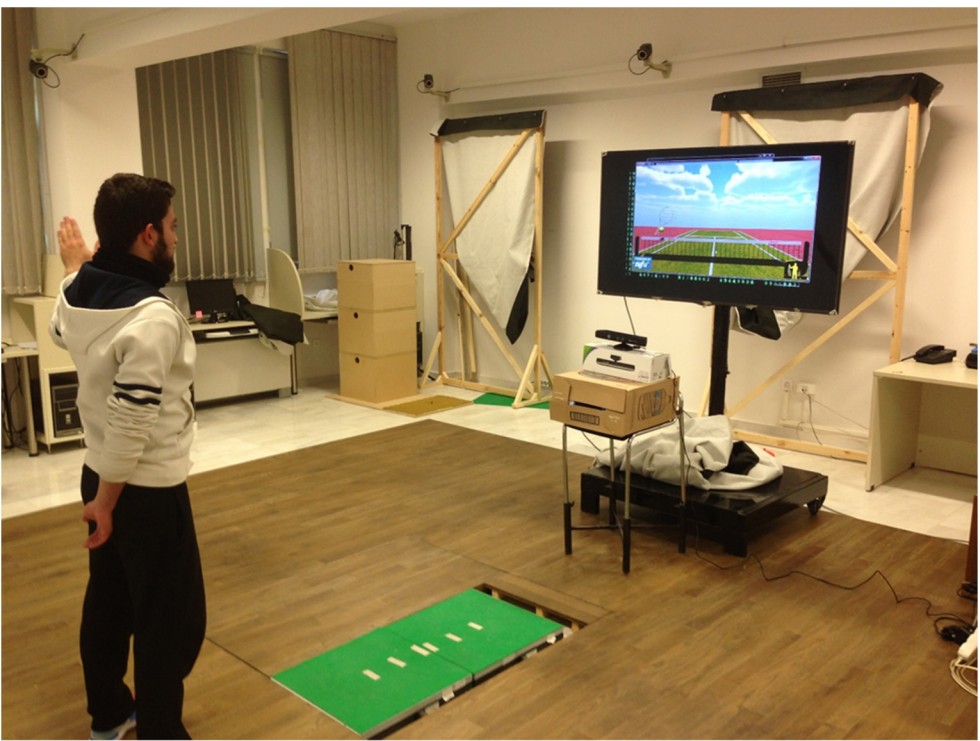

**Figure 3.** Gameplay of Tennis Attack.

## 5. Results

The data analysis based on the USE questionnaire explores the perceived usability, namely, how useful users found the total solution. With respect to the reaction time, measurements were divided in two: (a) Measuring the reaction time in real life, and (b) measuring the reaction time in the game. The reaction time in real life was measured with a pre-post method, whereas the in-game reaction time was measured with a simple procedure. We compared the average reaction time of the first session (average of the first five rounds) and compared them with the average of the last training session. Notably, SPSS version 21 was used for the statistical analysis. The evaluation results are presented in the section below.

### 5.1. USE Questionnaire

By performing the statistical analysis, we tried to determine whether Tennis Attack is user friendly to all users and to find any relevant difference between users with difficult backgrounds.

As it can be seen in Figure 4, the average results of USE questionnaire are very promising. All four axes are above 3.5 with Satisfaction and Ease of Use at almost 5 and Ease of Learning at almost 5.5. Sports Science students, as experts, were more receptive to the game's Usefulness.

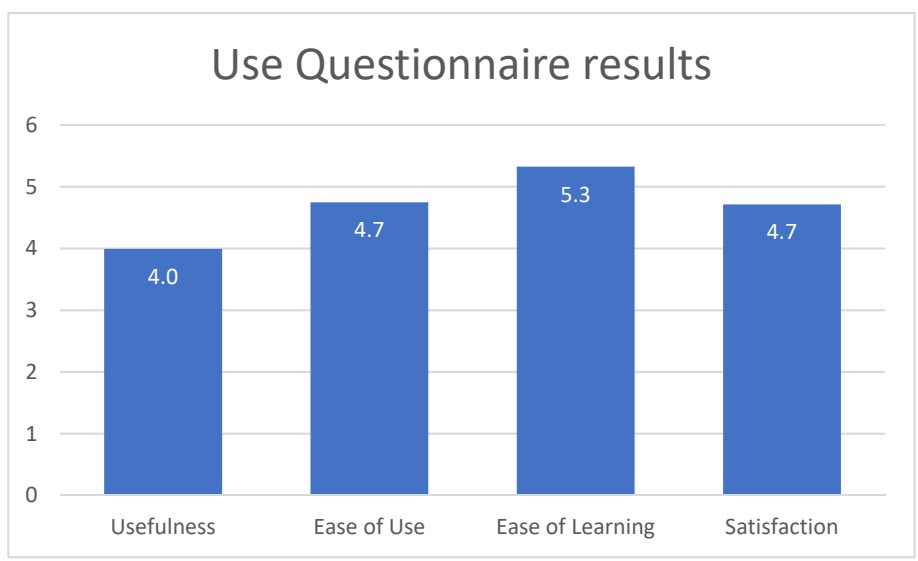

**Figure 4.** USE questionnaire results.

Additionally, as it can be seen in Table 1, there are differences between the mean values of Usefulness, Ease of Use, and Satisfaction between the two groups. To further investigate this and find the significance, we conducted a one-way ANOVA to all axes and tested the hypothesis that the user is a sports science student or a computing student.

**Table 1.** USE questionnaire results.

| | Mean | | | Std. Deviation | | |
|---|---|---|---|---|---|---|
| | **Sports Science Student or Computing Student** | | | **Sports Science Student or Computing Student** | | |
| | **Sports Science Student** | **Computing Student** | **Total** | **Sports Science Student** | **Computing Student** | **Total** |
| Usefulness | 4.3 | 3.7 | 4.02 | 0.452 | 0.534 | 0.575 |
| Ease of Use | 5.00 | 4.59 | 4.78 | 0.589 | 0.644 | 0.645 |
| Ease of Learning | 5.48 | 5.2 | 5.36 | 0.473 | 0.380 | 0.434 |
| Satisfaction | 4.94 | 4.59 | 4.74 | 0.482 | 0.564 | 0.562 |

The results of one-way ANOVA proved that there is a statistically significant difference in the axes (Usefulness, Ease of Use, and Satisfaction). Based on the student's expertise, there was a statistically significant difference in Usefulness, Ease of Use, and Satisfaction ($F_{(8, 27)} = 3.372$, $p < 0.0005$; Wilks' $\Lambda = 0.500$; partial $\eta^2 = 0.500$).

*5.2. Real-Life Reaction Time*

As previously described, to measure the real-life SRT (one stimuli—one response), we used a simple verified online click test (Table 2).

**Table 2.** Real-life reaction time of the participants.

| | N | Minimum | Maximum | Mean | Std. Deviation |
|---|---|---|---|---|---|
| Pre-reaction time | 60 | 0.214 | 0.398 | 0.32218 | 0.048712 |
| Post-reaction time | 60 | 0.217 | 0.385 | 0.30020 | 0.039938 |

As seen in Table 3, the results are impressive since there is a significant difference (sig < 0.005) between the pre- and post-measurement. To test the significant difference, we conducted a paired samples *t*-test. The mean improvement in users was 200 ms (Table 3).

Additionally, the improvement is not gender related. Both male and female students had the same improvement as there were no statistically significant differences between their mean values.

**Table 3.** Pre- and post-evaluation of statistical significance for improvement in the real-life reaction time.

| | | **Paired Differences** | | | | | | | |
| | | **Mean** | **Std. Deviation** | **Std. Error Mean** | **95% Confidence Interval of the Difference** | | *t* | **df** | **Sig. (Two-Tailed)** |
| | | | | | **Lower** | **Upper** | | | |
| Pair 1 | Pre-reaction time– Post-reaction time | 0.022 | 0.016 | 0.002 | 0.017 | 0.026 | 10.409 | 59 | 0.000 |

To investigate the hypothesis that sports science students would significantly improve their reaction time compared to computer science students, we conducted an independent samples *t*-test. This study found that the improvement in reaction time of sports science students (0.289 ± 0.037 s) is not significantly larger than the computer science students (0.307 ± 0.040 s); $t (58) = 1.755$, $p = 0.085 > 0.005$ (Table 4).

**Table 4.** Independent samples *t*-test (computer science students and sports science students).

| | | **Levene's Test for Equality of Variances** | | **t-Test for Equality of Means** | | | | | | |
| | | **F** | **Sig.** | *t* | **df** | **Sig. (Two-Tailed)** | **Mean Difference** | **Std. Error Difference** | **95% Confidence Interval of the Difference** | |
| | | | | | | | | | **Lower** | **Upper** |
| Post-reaction time | Equal variances assumed | 0.002 | 0.967 | 1.755 | 58 | 0.085 | 0.018 | 0.010 | −0.003 | 0.039 |
| | Equal variances not assumed | | | 1.772 | 53.600 | 0.082 | 0.018 | 0.010 | −0.002 | 0.039 |

Furthermore, another parameter that required investigation was whether being a gamer will have an impact on the reaction time improvement. As it can been seen in Table 5, the reaction time improvement in non-gamers (0.301 ± 0.038 s) is not significantly larger compared to gamers (0.298 ± 0.043 s); $t (58) = 0.202$, $p = 0.725 > 0.005$.

**Table 5.** Independent samples *t*-test (gamers and non-gamers).

| | | **Levene's Test for Equality of Variances** | | **t-Test for Equality of Means** | | | | | | |
| | | **F** | **Sig.** | *t* | **df** | **Sig. (Two-Tailed)** | **Mean Difference** | **Std. Error Difference** | **95% Confidence Interval of the Difference** | |
| | | | | | | | | | **Lower** | **Upper** |
| Post-reaction time | Equal variances assumed | 0.125 | 0.725 | 0.202 | 58 | 0.841 | 0.002 | 0.011 | −0.0191 | 0.023 |
| | Equal variances not assumed | | | 0.197 | 45.234 | 0.845 | 0.002 | 0.011 | −0.0198 | 0.0240 |

### 5.3. In-Game Reaction Time

Herein, we wanted to measure the improvement in the in-game reaction time. Additionally, we determined whether there is a significant change and whether any differences are dependent on the different groups that we defined previously. In Table 6, we can see the minimum and maximum values, the mean value, and the standard deviation of the participants before and after the training session.

**Table 6.** In-game reaction time of the participants.

|  | N | Minimum | Maximum | Mean | Std. Deviation |
|---|---|---|---|---|---|
| Pre-training Session | 60 | 0.568 | 1.387 | 0.965 | 0.190 |
| Post-training Session | 60 | 0.602 | 1.235 | 0.820 | 0.127 |

As we can see in Table 6, there is an improvement in the mean reaction times of the players before and after the training session (0.145 s). To investigate the hypothesis that this difference is statistically significant, we conducted a paired samples *t*-test.

Furthermore, as in real life, we investigated the hypothesis that there might be a significant difference between the groups that we have created (different expertise and gaming experience). In addition, the difference in mean times was not statistically significant (sig > 0.005) (Table 7).

**Table 7.** Evaluation of statistical significance for improvement in the in-game reaction time.

|  |  | Paired Differences | | | | | *t* | df | Sig. (Two-Tailed) |
|---|---|---|---|---|---|---|---|---|---|
|  |  | Mean | Std. Deviation | Std. Error Mean | 95% Confidence Interval of the Difference | | | | |
|  |  |  |  |  | Lower | Upper | | | |
| Pair 1 | Pre-training Session–Post-training Session | 0.144117 | 0.109806 | 0.014176 | 0.115751 | 0.172483 | 10.166 | 59 | 0.000 |

## 6. Discussion

This article is an early attempt to investigate the possibility that an active video game using NUIs can improve the simple reaction time.

The data collected and analyzed in the previous section have produced several interesting and impressive results regarding all the research questions raised. These results are presented in the following paragraphs.

**RQ1:** Does Tennis Attack improve the in-game reaction time of players? The statistical analysis presented in Section 5.3 proved that Tennis Attack can improve the reaction time of in-game players. As they train and learn how to use the game, they continuously get better. The reaction times of the players are not similar to the real world due to the limitations of technology. Some computers could add an additional 10–50 ms, while some modern TVs could add as much as 150 ms. This finding is in line with Letovsky [29] who stated that player's hand-eye coordination is very important for the human reaction time and its training is vital to improve the reaction time. Moreover, Greenfield et al. [30] reported that attentional training grants are important for improvement in overall visual monitoring performance.

**RQ2:** Does Tennis Attack improve the real-life reaction time of players? According to the presented statistical analysis (Section 5.2), the reaction time for most of the participants were significantly improved after the activity. Therefore, the improvement in the reaction time was not related to any previous experience with video games (RQ4) or their expertise (RQ3). As we have observed, the improvement was almost identical to all subcategories, which makes Tennis Attack a video game that anyone can use and benefit from. This finding is in line with Green et al. [31] and Trout et al. [32] who stated that video games enhance visual motor skills and improve the reaction time. Additionally, as Hulteen et al. [25] stated, some skills can be improved using AVGs.

**RQ3:** Is there any significant relationship between a player's expertise and their improvement? As previously shown, there is no significant difference between the mean times of users' real-life and in-game reaction times. Both groups can benefit from Tennis Attack. They equally improve in the game and as a result in real life, which allows them to transfer this improvement into their everyday experiences and even to sports activities. This finding is in line with the research by Bickmann et al. [33] who proved that traditional sportspeople, professional, and non-professional eSports players showed no differences in SRT improvement.

**RQ4:** Is there any significant relationship between a player's gaming experience and their improvement? As shown in RQ3, there is no significant difference between the two groups (pro-gamers and casual gamers). These results are similar to the findings by Green et al. [34], who stated that extensive training on an active video game equalizes the improvement in reaction time, make it beneficial for both experienced and casual gamers.

**RQ5:** Is Tennis Attack user friendly to all users? Based on the results presented in Section 5.1, Tennis Attack is designed in a way that experts, athletes, and amateur users can use it. The results proved that Tennis Attack is easy to use, easy to learn, and gives players a sense of satisfaction. Additionally, experts found it more useful than casual users, which is very important. These findings are in line with Blake's guidelines for designing NUIs [3], in which experts re-use specific domain motor skills and can understand the importance of this game.

## 7. Conclusions

The present study uses a newly developed active video game to improve reaction time and provides a training schedule. The proposed training schedule is designed for 4 weeks, with three sessions per week and five sets per session, which allows the players to benefit and actually improves their reaction time. The research evaluation findings have shown that the recommended application (Tennis Attack) and training procedure significantly improved the SRT of players both in real life and in game, as presented in RQ1,2. An expert, namely, the coach of the Tennis athletes from the Department of Physical Education and Sport Science, stated after the presentation of the results, that he considers the proposed training procedure useful for his athletes and easy to be deployed in their training program. Players were continuously getting better while they were playing the game and the results at the end demonstrated this improvement. In summary, it seems that NUIs and exergames can be combined to improve players' SRT when they are used with an appropriate training program.

Moreover, the study reveals that there is no significantly positive effect on users with experience in gaming or different expertise (RQ3,4). This was anticipated by the researchers due to a previous research [33]. This finding is important since Tennis Attack seems to be a tool that anyone can use and not only the experts. Additionally, the opinion of users was positive, whereby they were satisfied while playing the game and it was easy for them to learn and use.

## 8. Limitations

The findings of the present study include some limitations. First, the lack of space and equipment to train multiple players at the same time was a very important factor. As a result, it was not possible to carry out this research with a larger group of players. Another limitation is associated with the Microsoft Kinect camera, which only captures 60 frames per second. Therefore, sometimes, it could not capture the movement and the athlete had to start his training session from the beginning.

## 9. Future Work

As we have previously described, the results are very promising. Further research is required to investigate the possibility that Tennis Attack will be beneficial for younger players. In addition, it would be useful to investigate whether the improvement in reaction

time is different for different ages. Another interesting measurement is whether different sexes have different improvements in their reaction time.

To extend the results of this research to more groups, we could use Tennis Attack among older adults with moving disabilities. This will test the game's potential to help with rehabilitation.

Another important issue is to create another mode to measure and train players using the Choice Reaction Time, which evaluates general alertness and motor speed. It is a two-choice reaction time test, similar to the SRT task; however, the stimulus and response uncertainty are introduced by having two possible stimuli and responses.

Furthermore, based on the positive opinion of the coach of the Tennis athletes from the Department of Physical Education and Sport Science, a future step could be the deployment of the proposed training procedure to a larger scale to investigate the acceptance of the proposed procedure by a higher number of coaches in Tennis or in other sports.

**Author Contributions:** Conceptualization, N.P. and T.T.; methodology, N.P. and T.T.; software, N.P.; validation, N.P. and T.T.; formal analysis, N.P.; investigation, N.P.; resources, N.P.; data curation, N.P.; writing—original draft preparation, N.P.; writing—review and editing, T.T.; visualization, N.P.; supervision, T.T.; project administration, N.P. and T.T. All authors have read and agreed to the published version of the manuscript.

**Funding:** This research received no external funding.

**Institutional Review Board Statement:** The study was conducted in accordance with the Declaration of Helsinki. Ethical review and approval were waived for this study because the demographic data used and collected were anonymised.

**Informed Consent Statement:** Informed consent was obtained from all subjects involved in the study.

**Data Availability Statement:** Not applicable.

**Conflicts of Interest:** The authors declare no conflict of interest.

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
