# Peer review of "Tennis Attack: An Exergame Utilizing a Natural User Interface to Measure and Improve the Simple Reaction Time"

_applsci, doi:10.3390/app12199590_

Round 1

Reviewer 1 Report

The paper describes initial but interesting research concerning the empirical findings about the exergames. The authors formulated 5 research questions and performed the study involving 60 students. 

The abstract could be improved. It should provide information about the research method as well as present the results for all research questions. 

Since there is a gender division in sports games, it seems to me that in this case, it would also be possible to determine whether there is a difference in results by gender. Especially, when you provided the division in the description of the research groups.

The visual part of the paper could be greatly improved. I suggest providing a schema/illustration of the procedure described in Section 4.4 as well as the training schedule and the whole experiment. It would be much easier to grasp.

Moreover, in the case of USE Questionnaire results, maybe a box chart would better illustrate the data than tabular results. In addition, providing 5 digits after the decimal point is rather insignificant.

Minor issues:

- Instead of using "his/her", I suggest using singular they.

- The first table is not numbered: "Table t: USE Questionnaire results".

- Comparing Table t and Table 1, the usage of a dot (.) and comma (,) is inconsistent.

- "on Section 5" => "in Section 5", On Table 6" => "In Table 6", "on paragraph" => "in paragraph".

Author Response

Dear Reviewer,

we would like to thank you for your comments.

Please find attached our answers to all of them.

Regards

The aurthors

Reviewer 2 Report

Find here some comments to take into account for a better version: 

1. This paper advocates NUIs to improve motor skill and simple reaction time, but the conclusion section only emphasizes that it helps in simple reaction time.  Avoid confusion and put a better title according to the content/proposal of this paper. 

2. Thank you for providing a broader view In sections 1 and 2, In addition, in section 3 several research questions are raised to be answered by this research.  But please argue the conditions under which the proposed hypothesis was possible. 

3. According to the conclusion and the result related to the research questions. Does the research of this paper improve the reaction time for training through the use of exergames with NUIs? 

4. Restructure the conclusion indicating the valuation of the hypothesis, answer the previous question and, what objective(s) have been achieved.

Author Response

(The authors gave the same response as above.)

Round 2

Reviewer 1 Report

I would like to thank the authors for improving the paper according to the suggestions. I hope your work will be interesting to the journal readers.